# Quaternized Polyethersulfone (QPES) Membrane with Imidazole Functionalized Graphene Oxide (ImGO) for Alkaline Anion Exchange Fuel Cell Application

**Moshito Lethabo Mothupi [1,2] and Phumlani Fortune Msomi [1,2,*]**

1 Department of Chemical Science, University of Johannesburg, Johannesburg 2028, South Africa
2 Research Centre for Synthesis and Catalysis, University of Johannesburg, Johannesburg 2028, South Africa
* Correspondence: pmsomi@uj.ac.za; Tel.: +27-11-559-6313

**Abstract:** Membranes made up of composite materials have shown promising advantages for application in alkaline fuel cell anion exchange membranes. In this study, a general method is employed to improve the overall performance and properties of alkaline anion exchange membranes by making use of polyethersulfone quaternized with imidazolium groups and blended with imidazolium functionalized graphene oxide inorganic filler (ImGO). The inorganic filler blended with the polymer matrix yielded better ionic transport, with 73.2 mS·cm$^{-1}$ being the highest ion conductivity for the polymer membrane with 0.5% ImGO content, which is higher than that of the QPES parent material. The 0.5% ImGO content also showed better swelling ratio, water uptake, alkaline stability, ion exchange capacity and alkaline stability in comparison to other membranes. Furthermore, it also exhibited 130 mW·cm$^{-2}$ peak power.

**Keywords:** alkaline anion exchange membrane (AAEM); polyethersulfone (PES); graphene oxide (GO); imidazole; fuel cells; functionalization

## 1. Introduction

Alkaline anion exchange membrane fuel cells (AAEMFCs) have piqued the interest of researchers as a potential green and highly efficient source of energy [1,2]. To commercialize this type of fuel cell, enormous endeavors have been made in research so that a promising energy conversion device can be developed [3,4]. The work on AEMFCs mainly focuses on membranes as the electrolyte, historically the fuel cell, was made up of a liquid electrolyte. The formation of carbonate or bicarbonate ($CO_3^{2-}$ or $HCO_3^-$) in the liquid electrolyte, which occurs when hydroxide ions in the electrolyte react with $CO_2$ contaminants in the oxidant gas stream or other traces in the atmosphere, induced electrode degradation [5,6].

The main cause of the degraded performance of the fuel cell was that the carbonate or bicarbonate precipitate into large solid metal carbonate crystals, usually $Na_2CO_3$ or $K_2CO_3$, depending on the liquid electrolyte utilized, and fill the electrolyte and electrode pores, causing them to be blocked and leading to the destruction of the active layers [7]. The concentration of hydroxide ions in the electrolyte also becomes reduced, as well as the overall ionic conductivity of the fuel cell [8]. To prevent these phenomena, anion exchange membranes (AEM) consisting of a solid polymeric material have been developed in replacement of the liquid electrolyte. The AEM retains the electrocatalytic aspects of the alkaline fuel cell and $CO_2$ tolerance since there are no mobile cations that can result in carbonate or bicarbonate precipitate formation [7].

Solid polymeric membranes in alkaline fuel cells differ from those in proton exchange fuel cells [4]. The alkaline membrane in AEMFCs allows the transportation of $OH^-$, and the acidic membrane in PEMFCs allows the transportation of $H^+$; the alkaline fuel cell offers great advantages as compared to PEMFCs. These advantages include enhanced oxygen reduction catalysis due to the alkaline environment; this enables the use of a

less costly non-Pt catalyst or a PGM-free catalyst, making use of a wide range of fuels such as nitrogen-based fuels (ammonia, methanol, ethyl glycol or hydrazine) in addition to pure hydrogen, unlike PEMFCs, whereby other fuels rather than hydrogen can be employed. Since fluorinated raw materials are not used, a wider range of less expensive polymer chemistry is accessible. AEMFCs have extremely low or no fuel crossover, while PEMFCs have high fuel crossover; if the fuel passes over, the efficiency of the fuel cell decreases [9–12].

Although AAEMFCs are advantageous over commercialized PEMFCs, they still possess drawbacks with the membranes utilized. Since the membranes are synthesized by the introduction of halogen allyl groups onto the backbone or side chains of the polymer followed by the quaternization or amination by reaction with amine-based groups, which are susceptible to hydroxide attack via Hoffmann elimination and E1 elimination due to their insufficient stability, this leads to the reduced performance of the membrane since the membrane's conductivity, mechanical, and alkaline stabilities are compromised [12–15].

The alkaline anion exchange membrane with increased ionic channels for ion conductivity, strong mechanical, and chemical stability even under high alkaline temperature environments, possesses low cost, ease of fabrication, and zero electronic conductivity, that is, it must act as a barrier for electrons; no electrons should pass through the membrane, and it should provide optimal efficiency. Significant attempts have been made to enhance these properties, such as the exploitation of novel polymers, polymer–inorganic composite membranes, and polymer–polymer blends membranes. Some research has also focused on developing a new class of proton and anion exchange membrane based on electrospun fibers and composite nanofibers [15]. The most appealing of these approaches is the use of polymer–inorganic hybrid membranes (composite) with alkaline quaternary groups, such imidazolium, phosphonium, and guanidinium cations [15–18].

Quaternary imidazolium groups are more widely used than quaternary alkaline groups owing to the conjugated imidazole ring's resonance effect that lowers the positive charge density of the cations and weakens the interactions that exist within the hydroxide in alkaline environments. The presence of conjugated structures also helps in circumventing nucleophilic attack in the alkaline environment brought about by Hoffmann or $SN_2$ elimination. To date, several organic–inorganic membranes containing imidazolium have been synthesized, characterized, and tested for use in alkaline fuel cells [19,20].

Due to its 2D morphology, which provides both constant and long-distance ion transport network channels in the composite, graphene oxide (GO) has been recognized as one of the best fillers for producing polymer–inorganic composite membranes. Because of its close interfacial contact with the polymer matrix, it is easier to diffuse in a number of polymer matrices. It also has a lot of oxygen-containing groups on its surface, which gives it a lot of versatility and multifunctionality in terms of improving the efficiency of composite membranes [21–23]. Because of these properties, further research on polymer/GO composite membranes as anion exchange membranes (AEMs) is emerging.

Mao et al. produced macromolecular brushes by the vast functionalization of imidazolium on graphene oxide (GO) which were combined with polysulfone functionalized with imidazolium to form a composite membrane of polymer–inorganic materials. The findings showed that functionalized quaternary imidazolium groups were extremely effective in facilitating hydroxide conduction, membrane swelling, uptake of water, and ion exchange conductivity [24].

Yang et al. synthesized varying membranes of poly (2,6-dimethyl-1,4-phenylene oxide)/PPO quaternized using imidazole and which were blended with graphene oxide functionalized with an ionic liquid (IL-GO) [25]. The membranes outperformed the imidazolium functionalized PPO membrane regarding ionic conductivity and chemical and mechanical stability. The highest ionic conductivity was measured at 78.5 mS·cm$^{-1}$ at 80 °C given by 0.5 $w/w$ % IL-GO content in the membrane. The increased alkaline stability was due to the addition of IL-GO, which inhibits the passage of the molecular chains, decreases the free volume of the AEMs, and therefore prevents the nucleophilic displacement of

$OH^-$. The highest power density was obtained at 136 mW·cm$^2$ with a current density of 300 mA·cm$^{-2}$.

Liu et al. developed anion exchange membranes made up of quaternized mesoporous silica nanoparticles and quaternized polysulfone; the mesoporous silica was chosen as a filler due its high surface regions and broad open channels which permits for high chemical and thermal stability. The results obtained showed that the membrane performance was increased by the incorporation of quaternized mesoporous silica in comparison to pure quaternized polysulfone membrane. The membranes which had 20% of the quaternized mesoporous silica showed improved results in the ion exchange capacity and resistance to swelling [26].

To obtain improved hydroxide conductivity, Li et al. produced a combination of graphene oxide and poly (ether ether ketone)/(PEEK) membranes with both GO and PEEK functionalized with imidazolium; ImGO integration increased the membrane ion exchange capacity to 2.59 mmolg$^{-1}$. Most imidazolium groups found on the ImPEEK and ImGO created an endless hydrogen bonded network with absorbed water leading to an enhanced hydroxide conductivity of 0.14 mS·cm$^{-1}$ at 70 °C with the membrane containing 4 wt% ImGO content. This conductivity was 68.7% higher, and a power output of 50.04 mW·cm$^{-2}$ was obtained at a temperature of 50 °C [27].

These studies and others have contributed significantly to the use of GO as a filler for fuel cell applications, but there is still no commercial membrane that has been proposed, indicating that the fabricated membranes are still not fully adequate. The above studies and others have never showed a composite membrane of polyethersulfone and GO both functionalized with the same imidazole cation. Having the same cation group in the membrane and filler will result in a membrane with many ion conductivity sites, and thereby results in enhanced ion conductivity, and the stability of imidazole in both membrane and filler will result in a membrane which is not prone to fast membrane degradation. These are the two most important enhancements required if we are to realize a commercially viable membrane for fuel cell application.

In the present work, GO is used as an inorganic filler modified by imidazolium groups, and polyethersulfone is chosen as the polymer matrix due to its advantages, including stability over a wide range of temperature, rigidity, and resistance to inorganic acids and bases (chemical stability) due to the sulfone group, ether linkages, and good structural stability due to the aromatic rings. Imidazolium is used as the quaternary agent due to the already above-mentioned properties. The polymer is modified by chloromethylation followed by quaternization with imidazolium. The functionalized GO and polyethersulfone were then blended to form polymer–inorganic composite membranes. This work further indicates the enhancement of anion exchange membrane basic properties using functionalized graphene oxide. This combination of composites, from the best of our knowledge, is the first time being reported and evaluated in the application of methanol alkaline fuel cells.

## 2. Materials and Methods

### 2.1. Materials

Polyethersulfone (PES), dichloromethane, paraformaldehyde, Tin chloride (SnCl$_4$), chloromethyl silane ((CH$_3$)$_3$SiCl), ethanol, methanol, 1-methy-2-pyrrolidone (NMP), graphite, hydrochloric acid (HCl), dimethylformamide (DMF), potassium permanganate (KMnO$_4$), 98% sulfuric acid (H$_2$SO$_4$), hydrogen peroxide (H$_2$O$_2$), 1-[Bis(dimethylamino)methylene]-1H-1,2,3-triazolo [4,5-b]pyridinium-3-oxid-hexafluorophosphate (HATU), sodium hydroxide (NaOH), potassium hydroxide (KOH), and 2-methylimidazole were all purchased from Sigma-Aldrich, Johannesburg, South Africa and utilized in their pure form.

### 2.2. Synthesis of Graphene Oxide (GO)

A previously reported method was employed to synthesize: a mixture of graphite and KMnO$_4$ was treated with a 9:1 H$_2$SO$_4$/H$_3$PO$_4$ ratio. The reaction was left to heat under constant stirring at 50 °C for 12 h followed by cooling at room temperature and drained

into ice with 30% $H_2O_2$. The mixture obtained was then washed with distilled water, HCl, ethanol, and ether. The solid obtained was then oven dried [28].

### 2.3. Functionalization of GO

To quaternize GO with 2-methylimidazole, sonication and mechanical stirring were used to thoroughly disperse 200 mg of GO in 100 mL of DMF for 1 h at 30 °C. This was followed by the addition of 300 mL of 2-methylimidazole (Im) to the suspension for 1 h while being sonicated and mechanically stirred. Following that, a coupling agent (HATU) of 10 mg was added to the reaction mixture, followed by transfer to a water bath (60 °C) with stirring and reflux condensation for 6 h to complete the reaction [27–30].

### 2.4. Chloromethylation of PES

An amount of 1 g PES was dissolved in 150 mL of dichloromethane for effective chloromethylation drop-wise. Additions of paraformaldehyde, tin chloride, and chloromethyl-silane were made to the dissolved PES, followed by mechanical stirring at 50 °C for 24 h. Prior to precipitation with methanol, the reaction mixture was left to cool to room temperature. The received precipitate was cleaned several times using deionized water [31,32].

### 2.5. Fabrication of Quaternized Polyethersulfone Blended ImGO Membranes

The phase casting inversion process was used to make the composite membranes. After sonicating the ImGO in varying ratios for 1 h in 1-methyl-2-pyrrolidone (NMP), chloromethylated PES as well as imidazole were poured separately in the reaction vessels and left to continuous mechanical stirring for 24 h at room temperature. After obtaining a homogeneous solution, the solution was placed in a desiccator to remove any air bubbles. A glass plate and casting knife of 50 μm were used for casting the solution, and the membranes were precipitated in water. The membranes were immersed in an imidazole solution to allow for complete quaternization and the creation of a QPES/ImGO composite membrane [31,32].

### 3. Characterization and Measurements

Magritek Spinsolve 60 carbon benchtop 1H and carbon NMR spectrometer was employed in acquiring PES, CMPES, and QPES [1]H NMR spectra using deuterated dimethyl sulfoxide as the solvent. The FTIR spectrums of GO, ImGO, PES, CMPES, and QPES were obtained by the Spectrum 100 FT-IR Spectrometer from Perkin Elmer with a horizontal ATR system. Thermo Scientific DXR2 Smart-Raman Spectrometer was used for the characterization of graphene oxide (GO) and imidazolium functionalized graphene oxide (ImGO). The surface morphological features of the GO, ImGO, and all the membranes (QPES/ImGO-x%) were obtained by using the TESCAN VEGA3 Scanning Electron Microscope (SEM) instrument.

To evaluate the water absorption ability of the membranes, their water uptake was measured by soaking the membranes (2 × 2 cm) in deionized water at varying temperatures of 25 °C, 60 °C, and 80 °C for 24 h, followed by drying to a constant dry weight ($M_d$). The following Equation (1) [17–19] was used to calculate the water uptake.

$$WU = \frac{M_w - M_d}{M_d} \times 100\% \tag{1}$$

where $M_w$ and $M_d$ are the mass of the water swollen and dry membrane, respectively. The surface hydrophilicity of the PES, QPES, and the QPES/x–ImGO membranes was obtained with DATA Physics optical contact angle. The sessile drop measuring procedure was used to determine the optical contact angle. Using a Gilmont syringe, a 2 μL globule of deionized water was deposited on a flat membrane surface, then the contact angle (θ) between the flat membrane and water dropped on it was measured. This was performed using the SCA20 version 4.1.12 build 1019 software.

Ion exchange potential (IEC) is a relevant membrane property which estimates the availability of the quantity of anion replacement groups of the membrane. The design of endless channels that enable ion transport in the membranes necessitates high IEC. To determine the IEC, the fabricated membranes were individually immersed for 24 h in 40 mL HCl of 1 M followed by titration with 1 M NaOH until phenolphthalein endpoint was attained. To calculate the number of ions exchanged, the following Equation (2) [17–19] was used:

$$IEC = \frac{(V_{acid} \times C_{acid}) - (V_{base} \times C_{base})}{M_d} \qquad (2)$$

The moles of HCl are subtracted from those of NaOH, and the value obtained is divided by mass of the dry membrane ($M_d$) to obtain IEC.

Alkaline stability measures how long the membrane anionic conductivity can change when exposed to high pH condition at varying time intervals. The alkaline stability was examined by submerging the membranes in a solution of 2 M NaOH at 25 °C, then measured by examining changes in the anionic conductivity. The membranes were soaked in the basic solution for 7 days, and the anion conductivity was measured and collected for 5 days. The membranes' hydroxide conductivity was assessed utilizing a four-probe alternating current (AC) electrochemical impedance spectroscopy (EIS). After allowing the membranes to soak in deionized water overnight, the membranes were tested in a cell at various AC current for the collection of impedance data. The membrane ionic resistance ($R_m$) is determined using nonlinear least squares regression analysis; the conductivity can then be determined by means of the following Equation (3) [17–19]:

$$\delta = \frac{L}{R_m \times A} \qquad (3)$$

The distance between the electrodes in centimeters is denoted by L, and the cross-sectional surface/area of the dry membrane is denoted by in $cm^2$.

The fuel cell performance efficiency was tested by using a DMFC. Firstly, the membrane electrode assembly (MEA) was assembled by using the best performing membrane in the study. Platinum on carbon cloths (purchased at fuel cell store, TX, USA) were used at the cathode and anode for the test, and the MEA was constructed with no hot pressing. A mixture of 2 M methanol and 2 M KOH was allowed to flow from the anode side through the cell to carry out the testing, and cell potential was measured galvanostatically as a function of current density.

## 4. Results and Discussion

### 4.1. Subsection Functionalization of Graphene Oxide (GO)

FTIR was employed to verify successful GO synthesis and modification. Figure 1 shows the FTIR images of GO and ImGO. The distinguishing peaks of virgin GO showed approximately at 3416 $cm^{-1}$ attributed to the stretch of the -OH bond, showing the presence of hydroxyl groups; the peak at 1734 $cm^{-1}$ arose due to vibrational stretch of the carbonyl (C=O); the C=C vibrational stretching of the aromatic ring results in a peak at 1617 $cm^{-1}$; and the O-H bending vibrations of the carboxyl group result in the observation of a peak at 1387 $cm^{-1}$. At 1204 $cm^{-1}$ is the peak for the epoxy's C-O-C vibrational stretching, while that of the alkoxy is observed at 1045 $cm^{-1}$; these are consistent to the previously reported literature [33,34]. Compared to pristine GO, the ImGO showed the existence of a novel characterization peak visible at 3162 $cm^{-1}$ owing to N-H. It is visible that some of the -OH groups that were attached to the GO were replaced. The C=O peak disappeared and there was an increase and shift in intensity of the O-H vibration from the carboxyl group from 1387 $cm^{-1}$ to 1405 $cm^{-1}$, indicating that there was an addition of C=N stretching vibration found on the imidazole ring [27].

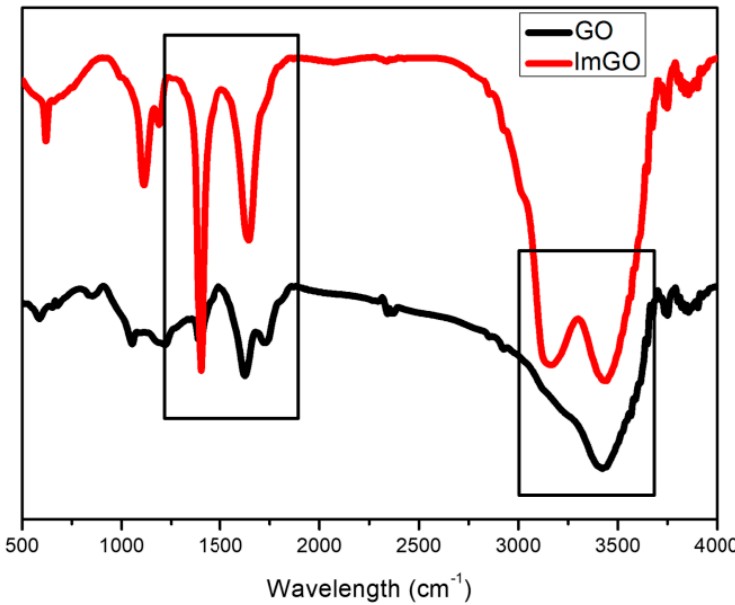

**Figure 1.** FTIR graphs of GO and ImGO.

To further clarify the synthesis and modification of GO by 2-methylimidazole, Raman spectroscopy was employed, as shown in Figure 2. Two prominent characteristic peaks are evidently visible at approximately 1356 cm$^{-1}$ and 1591 cm$^{-1}$, representing the D and G band. These bands signify the vibration of the sp$^3$ carbon atoms from the functional groups and the in-plane vibrations of the sp$^2$ carbon atoms, respectively [35]. The G band has a high sensitivity to strain effects in the sp$^2$ system and thus can be employed to explore modification on the surface of the graphene oxide [36]. In our case, the G band broadened and blue-shifted, indicating that the substitution of functional groups took place in the graphene oxide layer, and successful functionalization of the GO occurred. This can also be seen by the D and G band intensity ratio, which rose from 0.85 to 0.87 upon introduction of imidazolium, and the wavelength of the D peak in ImGO, which is higher than that of the G peak, indicating that, upon functionalization, the lattice structure of the GO was modified [37].

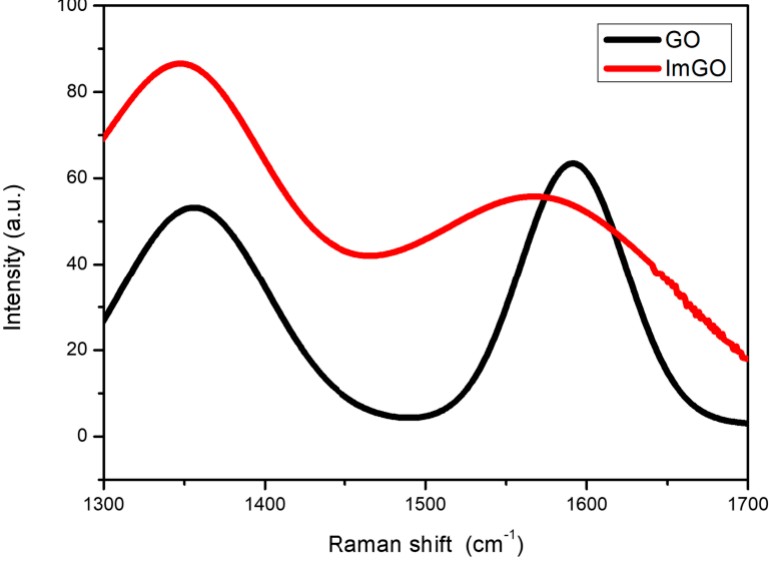

**Figure 2.** Raman spectra of GO and ImGO.

### 4.2. Functionalization of Polyethersulfone

The successful functionalization and quaternization of polyethersulfone were confirmed by FTIR as per Figure 3 and proton NMR as per Figure 4. In Figure 3, the structure of pure PES includes benzene rings, an ether bond, and a sulfone structure. The FTIR of PES shows a vibration from 710 cm$^{-1}$ to 1586 cm$^{-1}$, which is due to various skeletal and acoustic modes of vibrations of the functional groups on the PES structure. The peaks at 1581, 1480, and 1418 cm$^{-1}$ are due to vibrational modes of the aromatic ring, while the asymmetric and symmetric vibrational bands of the sulfonyl group showed at 1289 and 1149 cm$^{-1}$, respectively. The vibration of the phenoxide bound to the aromatic ring caused the peak at 1245 cm$^{-1}$. The bending vibrations of the C-H group found in the aromatic ring appeared at 1099, 1015, and 998 cm$^{-1}$, respectively. The aromatic ring 1, 4 disubstituted C–H bending vibrations occurred at 858 and 835 cm$^{-1}$. The successful chloromethylation of the PES is confirmed by an additional peak, which occurred at 779 cm$^{-1}$ corresponding to the C-Cl in the CMPES [38,39]. After quaternization with 2-methylimidazole, the C-Cl stretching band disappeared and there were additional peaks on the FTIR spectra of QPES at 1690 cm$^{-1}$ due to C=N vibration on the imidazole, indicating that quaternary imidazole unit was successfully introduced to the polymer structure.

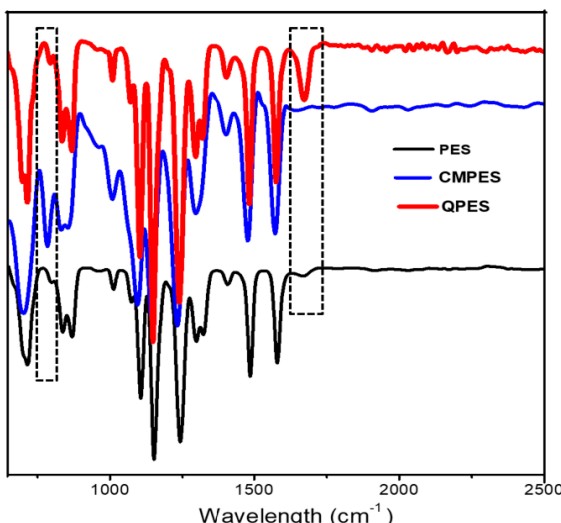

**Figure 3.** FTIR of PES, CMPES, and QPES.

To further confirm and verify the functionalization of PES, proton NMR was used, as shown in Figure 4a–c. The proton NMR of pure PES membranes shows two distinct doublet peaks between chemical shifts of 7 and 9 ppm, which are symmetrical to each other, and once chloromethylation takes place, there is an additional peak between 4 and 5 ppm, showing the protons attached to the carbon bearing and the chlorine group and indicating that the chloromethylation of the PES was successful. Upon quaternization, the peaks between 4 and 6 ppm became smaller due to the shielding brought about by the bulk group of the imidazole, and new peaks appeared at chemical shifts of 2–4 ppm and 6.5–8 ppm, which are due to the protons on the imidazole ring [24]. The FTIR and NMR agree that QPES was successfully fabricated.

### 4.3. Surface Morphology of GO and Membranes

The GO and membrane surface morphology was studied using Scanning Electron Microscopy (SEM). Figure 5A depicts the SEM of GO, which has ultrathin layers that are wrinkled at the edges and possess a kinked morphology. Figure 5B depicts the SEM of ImGO, which differs from that of GO in that the layers are not lined-up together, indicating that the GO has been changed [17].

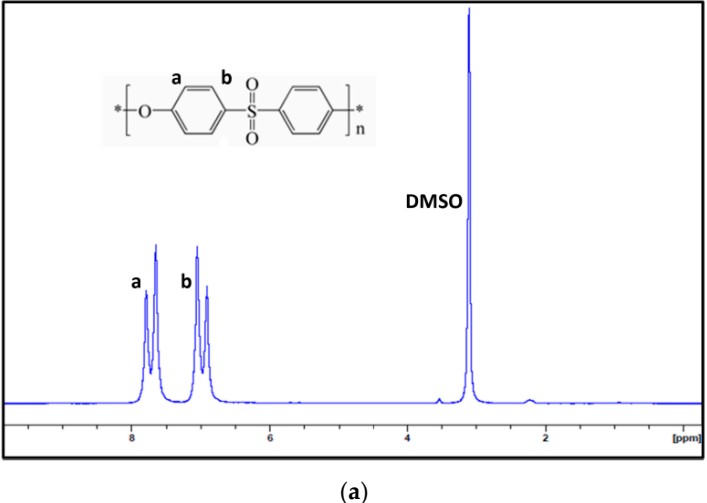

(**a**)

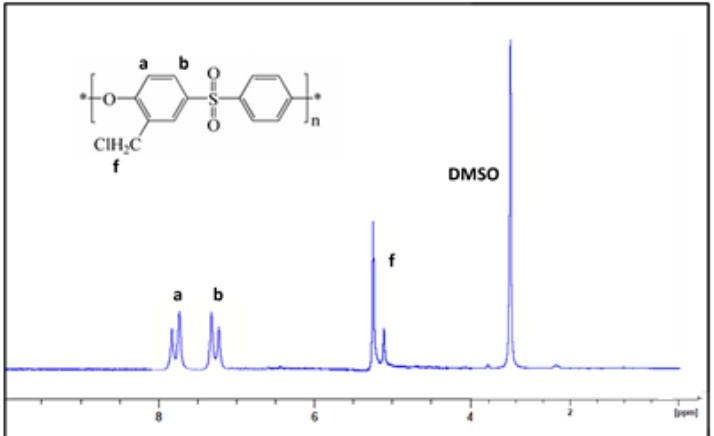

(**b**)

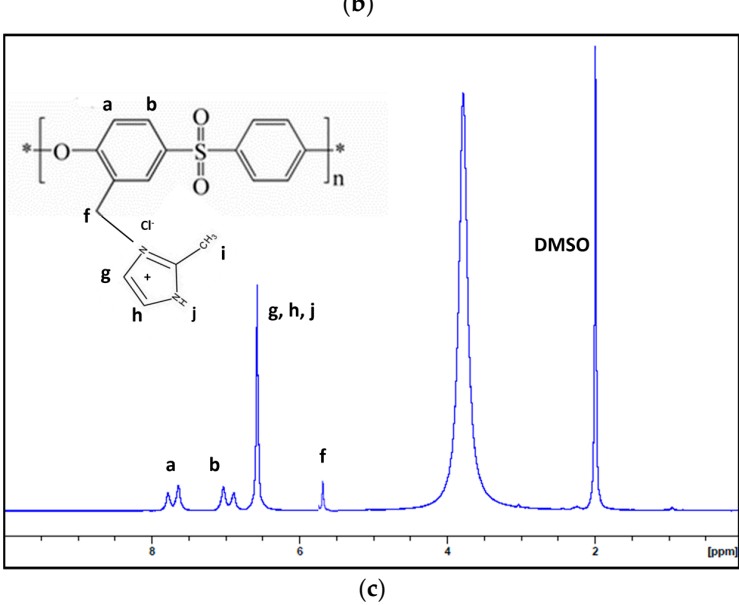

(**c**)

**Figure 4.** (**a**) [1]H NMR of PES. (**b**) [1]H NMR of CMPES. (**c**) [1]H NMR of CMPES.

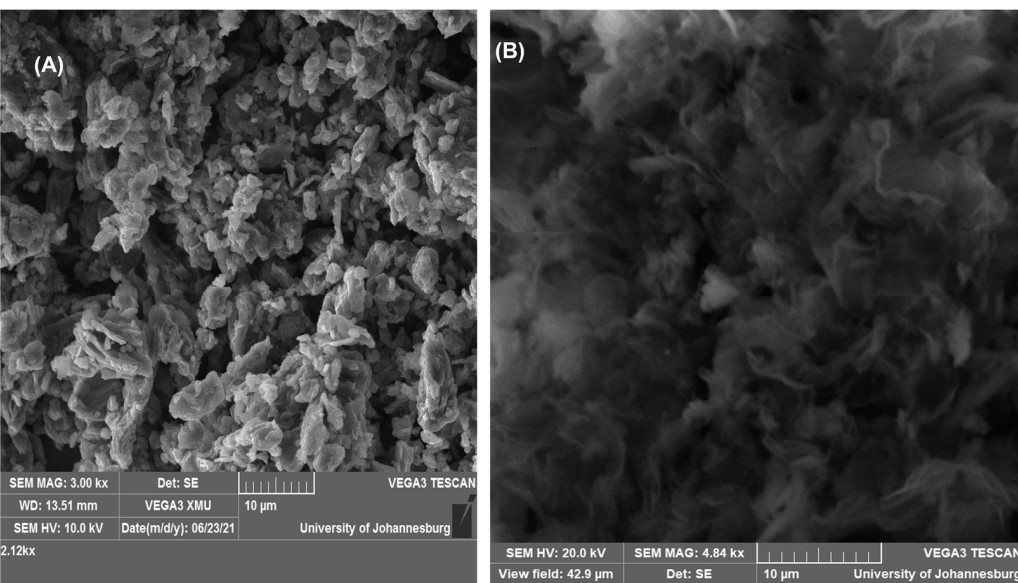

**Figure 5.** SEM images of (**A**) GO and (**B**) ImGO.

Figure 6A shows the SEM of pristine PES which has a smooth and even surface morphology and visible tiny pores which are not filled, but as soon as the PES membrane is quaternized with imidazolium, the morphology changes; the SEM image shows that the imidazolium was evenly distributed in the PES polymer matrix. During membrane pore modification, surface charge and bonding effects may take place leading to the etching of the membrane. The imidazolium appeared uniform and fixed on the membrane surface, indicating that further modification can be performed, as shown in Figure 6B [40].

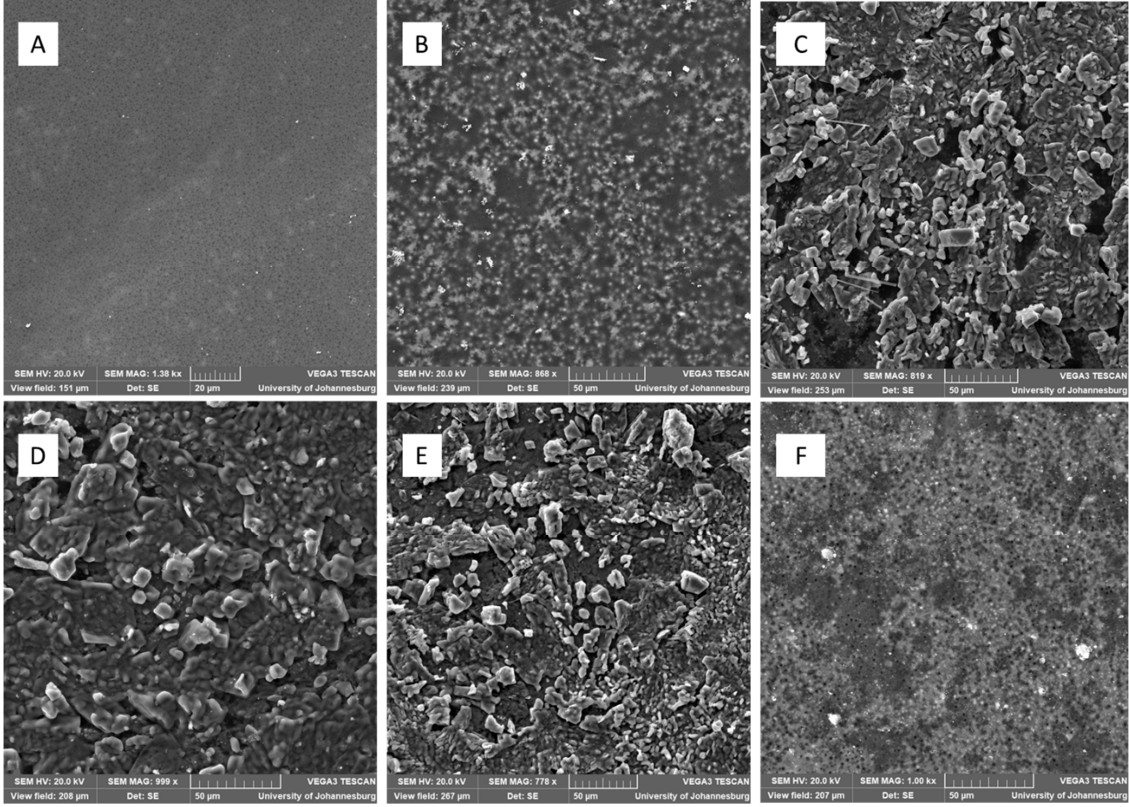

**Figure 6.** SEM images of (**A**) PES, (**B**) QPES, (**C**) QPES/ImGO-0.1%, (**D**) QPES/ImGO-0.3%, (**E**) QPES/ImGO-0.5%, and (**F**) QPES/ImGO-1%.

SEM was also employed to understand the morphology of the membranes after the introduction of ImGO. The SEM images with 0.1, 0.3, and 0.5% ImGO showed dense appearance and reveals significant clusters; these irregular clusters of the ImGO were formed on the surface of the QPES, and several underlying channels were observed underneath the surface clusters of the ImGO. The clustering might be due to the ionic agglomerate effects of the two positively constituents, QPES and IMGO. These clusters and the underlying channels facilitate in the transportation of ions and aids in faster kinetics of hydroxide ion transportation as expected in an alkaline fuel cell membrane. The SEM morphologies of QPES and QPES/ImGO membranes correlates with the work of Thanganathan Uma [41]. Upon increasing the ImGO content by 1%, the membrane pore becomes visible, showing that the ImGO does not blend effectively with the polymer matrix, as shown in Figure 6F.

### 4.4. FTIR of ImGO and QPES/ImGO-x% Membranes

To further confirm the incorporation of ImGO onto the polymer, FTIR was used. As per Figure 7, both the PES and GO were attached with imidazole to be conductive, so these functional groups overlap in the FTIR. It can be observed that all the membranes show a shoulder peak which is associated with the ImGO and is absent on the QPES. This indicates that the ImGO was successfully blended with the QPES to form a QPES/ImGO composite membrane. The intense shoulder at a chemical shift of 1617 cm$^{-1}$ was attributed to the ImGO filler in the polymer matrix, indicating successful incorporation, which is where the C=C bond of the graphene structure is seen.

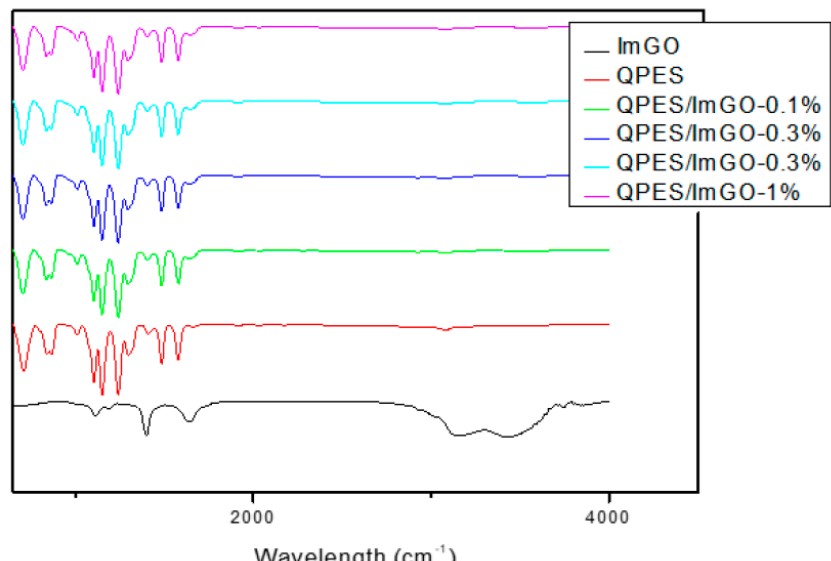

**Figure 7.** FTIR of ImGO and QPES/ImGO-x% membranes.

### 4.5. TGA Analysis of PES and QPES/ImGO-0.5%

TGA analysis was employed to examine the thermal stability of the PES and the best performing membranes, as seen in Figure 8. They both represented three steps of thermal degradation, with the first step happening in the range of 100 and 195 °C due to evaporation of impurities, residual NMP solvent, and water owing to the hydrophilic existence of the quaternized imidazolium and GO [42]. The second stage of thermal degradation occurs between 200 and 400 °C due to quaternary imidazole degradation on the backbone of the polymer [17]. The third step occurs at temperatures above 400 °C which leads to membrane weight loss caused by the thermal degradation of polymer chains and backbones. The QPES/ImGO-0.5% membrane was seen to exhibit similar degradation steps as the PES membrane; the result indicates that the ImGO was compatible with the polymer matrix and has the same degradation pattern. This further indicates that the QPES/ImGO membrane

can be used for low-temperature fuel cells which operate at 25 to 80 °C as per the aim of this study [43].

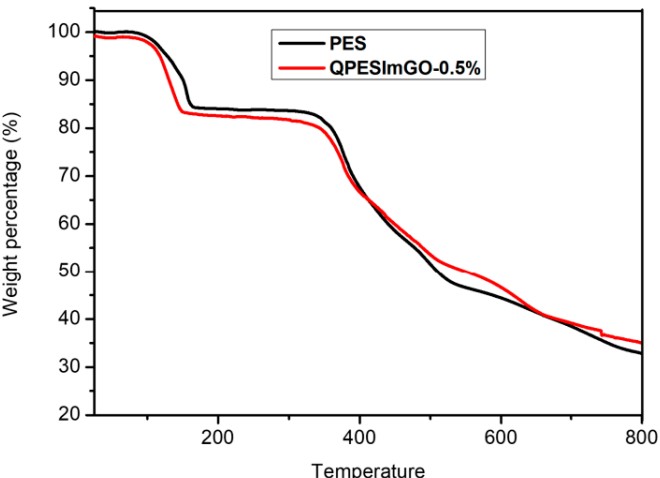

**Figure 8.** TGA of PES and QPES/ImGO-0.5%.

## 4.6. Membrane Evaluation

### 4.6.1. Water Uptake and Contact Angle

The membranes' water absorption is an important factor to consider as it influences both the ion exchange potential value and the mechanical properties of the membranes. At moderate WU values, the IEC and hydroxide conductivity improve; however, excessive water absorption by the membrane causes excessive harm due to poor mechanical properties and is not ideal for fuel cell use [44–46]. Figure 9 depicts the water absorption of the membranes at various temperatures and it is visible that, with temperature increment, the membranes' water uptake increased. The PES membranes showed an increase from 12.82% at 25 °C to 133.5% at 80 °C for PES. This is because high temperatures favored water plasticization and polymer relaxation, leading to an enhanced WU [47]. Comparison between the QPES and QPES/ImGO-x% revealed that an increase in the wt%of the ImGO increased the WU of the membranes. This is due to the hydrophilic nature that the imidazolium functionalized graphene oxide possesses; thus, the membranes with 0.5 wt% ImGO showed the highest WU.

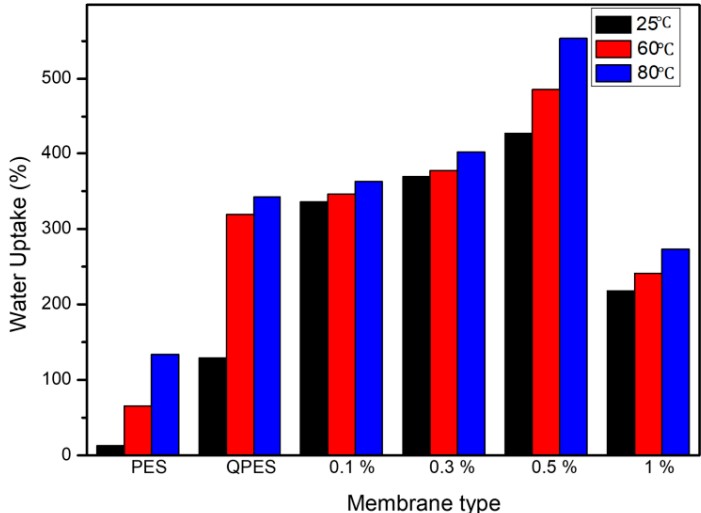

**Figure 9.** Water uptake of the membranes at different temperatures.

The 1 wt% ImGO showed a lower WU as compared to the rest of the ImGO functionalized membranes since the ImGO did not disperse well in the polymer matrix. The WU of the membranes is also linked to the amount of the imidazolium functional groups in the membrane, so the low hydrophilicity of this ImGO membrane as compared to 0.1, 0.3, and 0.5 wt% ImGO can be due to the membrane possessing low imidazole functionalities caused by poor dispersion.

The WU results presented are in correspondence with the contact angle results shown in Figure 10, whereby the membranes modified imidazolium and ImGO or which are properly dispersed with ImGO show a higher hydrophilicity compared to the control membrane, which in this case is PES. Based on Figure 9, the composite membrane with 0.5% ImGO shows better properties in terms of water uptake and contact angle.

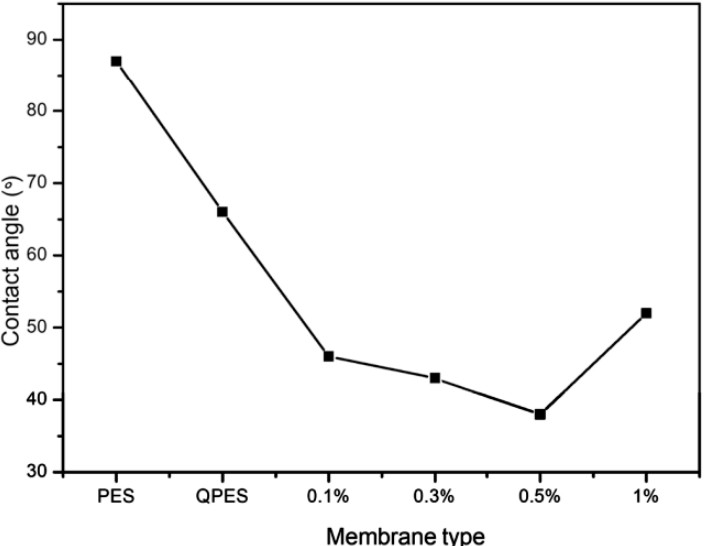

**Figure 10.** Contact angle of the membranes.

4.6.2. Ion Exchange Capacity (IEC) and Ion Conductivity (IC)

The membranes' IEC values are seen in Figure 11. The increase in the content of ImGO from 0% to 0.5% led to an increase in the IEC from 1.85 mmolg$^{-1}$ to 3.20 mmolg$^{-1}$, and the increase of ImGO content on the polymer matrix led to an increase in the quaternary imidazolium group, which increased the IEC of the membranes. When comparing the composites (membranes with ImGO), the lowest IEC is seen with the membranes containing 1% ImGO, and this can be attributed to agglomeration and poor dispersion due to a high amount of ImGO in the membrane matrix, thereby resulting in fewer ionic channels for ion transport. The 0.5% ImGO content membrane exhibited the highest IEC of 3.20 mmolg$^{-1}$. This is in accordance with the water uptake and contact angle.

The membranes' IC measurements show the anions' migration in the membrane pores, and an increased IC results in higher IC yields, higher membrane performance, and fuel cell output. The ion conductivity of the membranes was evaluated at 25 °C as per Table 1. The highest IC of 73.2 mS·cm$^{-1}$ was observed for the QPES/ImGO-0.5% membrane and the lowest was 35.8 mS·cm$^{-1}$ for the QPES membrane. All membranes that contained ImGO showed an improved IC compared to QPES, but a reduction in IC was observed with 1% ImGO content due to improper dispersion and loading. The IEC and ion conductivity (IC) are highly dependent on membrane water absorption and hydrophilicity. AEMs with high water absorption have been shown to have higher anion conductivity. The water inside the membrane serves as a hydroxide carrier, thus increasing the IEC and IC [48]. Table 1 shows the ion conductivity of all membranes prepared in this work.

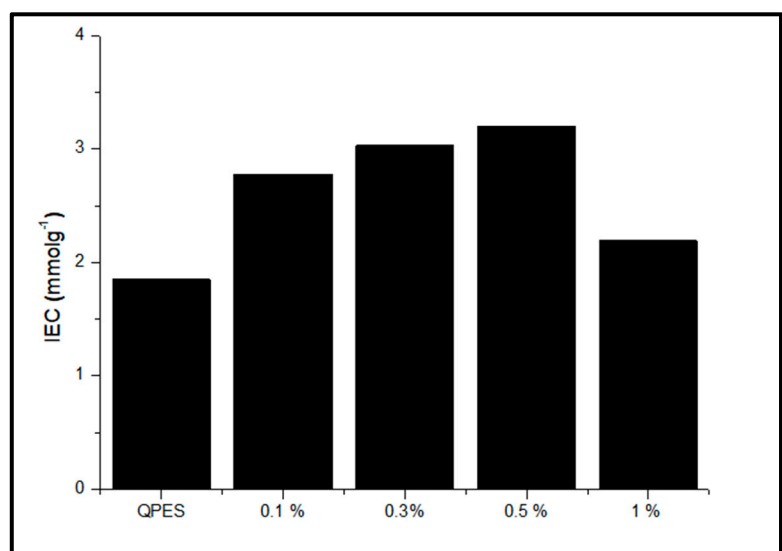

**Figure 11.** IEC of the membranes.

**Table 1.** Ion conductivity of the membranes.

| Membrane Type | IC (mS·cm$^{-1}$) |
|---|---|
| QPES | 35.8 |
| QPES/ImGO-0.1% | 56.8 |
| QPES/ImGO-0.3% | 47.1 |
| QPES/ImGO-0.5% | 73.2 |
| QPES/ImGO-1% | 50.2 |

Table 2 shows ion conductivity (IC) comparisons of the GO-based membranes in this study to those in previously reported studies. The membrane IC obtained in this study showed better results compared to IC found in the literature. The membranes QPES/ImGO-0.5% and QPES/ImGO-1% showed 73.2 and 50.2 mS·cm$^{-1}$ at room temperature, which is higher than the QPPO/PSF/2.0%GO and PBI/IL-GO-30 conducted at 80 °C. This occurred because the membranes in this study have both the polymer and the filler modified to form increased ion conducting channels in the membranes, unlike in QPPO/PSF/2.0%GO and PBI/IL-GO-30 membranes which have less modification in the membrane materials. So, an increase in temperature will be required for these membranes to have increased diffusion of the -OH ion within them, thus increasing membrane ion conductivity. However, the conductivity of PBI/IL-GO-30 is higher than that of QPES/ImGO-1%, since 1% ImGO is excessive for membranes in this study and causes undesirable stacking, forming sheets which are not able to fill the pores of the membranes which leads to reduced ionic pathways [48–50].

**Table 2.** The comparison of membranes' ionic conductivity in this work and the previously reported literature.

| Membrane Type | Temp (°C) | IC (mS·cm$^{-1}$) | Ref. |
|---|---|---|---|
| QPES/ImGO-0.5% | 25 | 73.2 | This study |
| QPES/ImGO-1% | 25 | 50.2 | This study |
| ImPSF/ImGO-0.2% | 80 | 35.89 | [24] |
| qPPO/C-1/G-1 | 25 | 112.89 | [49] |
| QPPO/PSF/2.0%GO | 80 | 63.67 | [17] |
| PBI/IL-GO-30 | 80 | 67.0 | [48] |

The ImPSF/ImGO-0.2% membrane has a low filler content as compared to the membranes in this study, leading to a reduced ion conductivity. As mentioned above, the temperature must be higher to obtain an increased diffusion of the ions within the membranes. At a low temperature of 30 °C, this membrane has a reduced ion conductivity of 22.02 mS·cm$^{-1}$. The qPPO/C-1/G-1 membrane has an enhanced ion conductivity of 112.89 mS·cm$^{-1}$ as compared to the ones in this study at a temperature of 25 °C. In the study, the membrane materials were all chemically modified with cation groups to form quaternized cellulose and quaternized graphene oxide hybrid nanofiller composite with quaternized polyphenylene oxide, thus leading to more cations present in the membrane, which form enhanced transport channels in the membrane translating to increased ionic conductivity. So, the membrane has more ionic transport channels than the membranes in this study.

### 4.6.3. Ion Exchange Capacity (IEC) and Ion Conductivity (IC)

The test of the membranes' alkaline stability over 160 h in 2 M NaOH solution carried out at varying time intervals is outlined in Figure 12. The QPES membrane was less stable under alkaline conditions as compared to all the membranes and showed a sharp decline in its ion conductivity, retaining only 30% of its initial ion conductivity after 160 h. It was followed by the QPES/ImGO-1% membrane, which reserved roughly 40% of its initial ion conductivity; the increased amount of the filler content in this membrane led to poor dispersion due to agglomeration thus reducing the alkaline stability. The QPES/ImGO-0.1% retained about 45% while the QPES/ImGO-0.3% retained about 50% of the initial ion conductivity. From this, it is clear that the correct amount of the filler content leads to an improvement in the membrane alkaline stability.

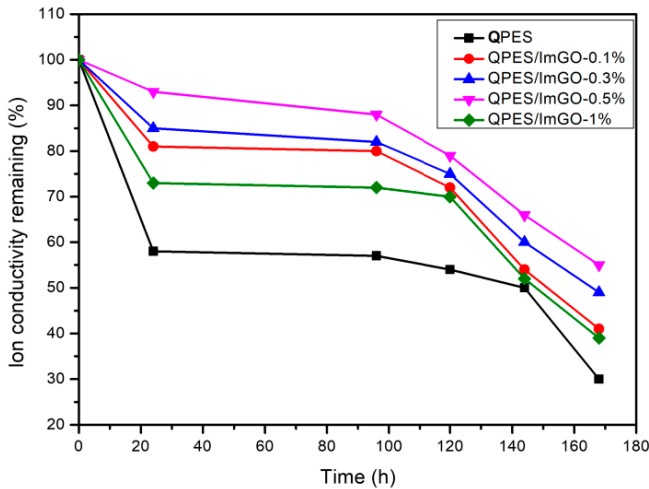

**Figure 12.** Membranes' alkaline stability.

The composite membrane of QPES/ImGO-0.5% was able to retain about 55% of its initial ionic conductivity after 160 h, displaying the highest alkaline stability. The QPES/ImGO membranes showed better alkaline stability as compared to the QPES membrane. This is attributed to the introduction of a vast amount of imidazolium cations on the QPES, leading to an enhancement in the alkaline stability by providing more ionic transport channels and steric hindrance, thereby reducing membrane degradation [10,48]. The alkaline stability also shows that the QPES/ImGO-0.5% composite membrane can be employed in alkaline fuel cells as an AEM since it retained the highest conductivity as compared to membranes in the study.

### 4.6.4. Fuel Cell Performance of QPES/ImGO-0.5% Composite Membrane

The QPES/ImGO-0.5% membrane, which performed better regarding ionic conductivity and stability in an alkaline setting during evaluation experiments in this work, was

employed to assemble a membrane electrode assembly (MEA) aimed at testing room temperature membrane performance using direct methanol fuel cell (DFMC). Figure 13 shows the cell polarization curves of this membrane. An amount of 250 mA·cm$^{-2}$ current density was obtained with a peak power density of 130 mW·cm$^{-2}$, and an open-circuit voltage of 0.99 V was obtained for the membrane at room temperature. These results correspond with the ion conductivity measurements showing that the migration of the anions within the membrane leads to increased conductivity due to the presence of quaternary imidazolium functionalities, which yields higher membrane performance and fuel cell output.

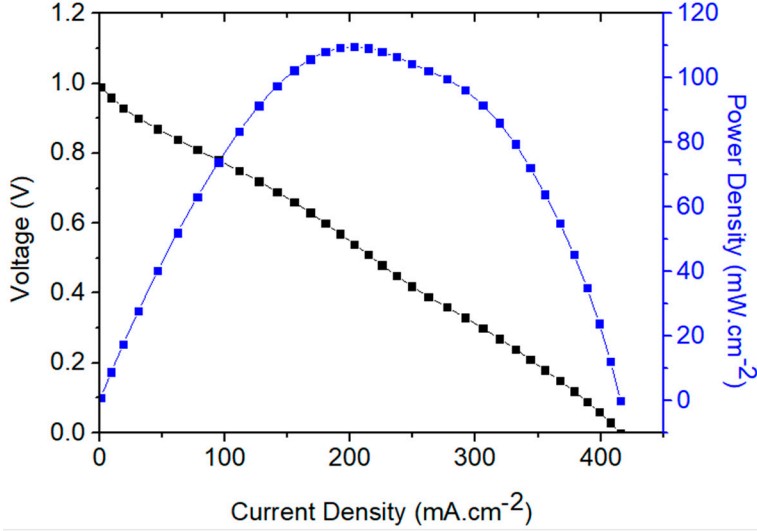

**Figure 13.** DMFC polarization curve of QPES/ImGO-0.5% composite membrane.

Table 3 compares the fuel cell output of this membrane to that previously recorded in the literature. The QPES/ImGO-0.5% results showed the highest power output as compared to those found in the literature, indicating that the membrane can be suitably applied as a polymeric membrane for fuel cell application. The high-power output is a direct result of having additional cation groups in both the membrane and the filler. This work significantly contributes to enhancing membrane properties for fuel cell application using GO as a filler.

**Table 3.** Comparison of QPES/ImGO-0.5% membrane power output in this study and other functionalized GO membranes found in the literature.

| Membrane Type | Peak Power Density (mW·cm$^{-2}$) | Temp (°C) | Ref. |
|---|---|---|---|
| ImPSF/ImGO-0.2 | 78.7 | 25 | [24] |
| QPPO/PSF/2%GO | 112 | 25 | [17] |
| QPES/ImGO-0.5% | 130 | 25 | This study |
| ImPEEK/ImGO-4 | 50.0 | 50 | [27] |
| QPPOQGO-2 | 5.20 | 60 | [49] |

## 5. Conclusions

In this work, new types of alkaline anion exchange membranes of 2-methyl imidazole functionalized GO and PES were successfully fabricated. The membranes were prepared by using a solution blending method followed by membrane casting with a phase inversion method. The resulting QPES/ImGO demonstrated improved ionic conductivity, ion exchange capacity, WU, and alkaline stability as compared to pristine QPES. The QPES/ImGO-0.5% was the best performing and showed high water uptake, IEC value of 3.20 mmolg$^{-1}$, IC of 73.2 mS·cm$^{-1}$, and an improved alkaline stability in comparison to virgin QPES. Furthermore, the power density of the QPES/ImGO-0.5% membrane was

130 mW·cm$^{-2}$ at room temperature. The vast amount of imidazolium groups on the ImGO surface forms a network of channels with reduced resistance for the transportation of ions, thus enhancing the overall performance. To the best of our knowledge, it is the first time a membrane (PES) and GO both functionalized with 2-methyl imidazole have been reported and applied in alkaline fuel cells. The fabricated membranes add to the body of knowledge of different composite membranes, which have a potential long-term application in alkaline fuel cells.

**Author Contributions:** Writing—original draft, M.L.M.; review and editing, resources, P.F.M. All authors have read and agreed to the published version of the manuscript.

**Funding:** This research was funded by National Research Foundation (NRF-South Africa-TTK 121930), National Nanoscience Postgraduate Teaching and Training Platform (NNPTTP) and the University of Johannesburg.

**Institutional Review Board Statement:** Not applicable.

**Informed Consent Statement:** Not applicable.

**Data Availability Statement:** This work resulted from a Masters in Nanoscience: Chemistry which was conducted by Ms. M.L. Mothupi under Dr. P.F. Msomi's supervision; she has submitted the final thesis to the University of Johannesburg. The thesis can be found in the University of Johannesburg's library with the link https://hdl.handle.net/10210/484141 (accessed 30 November 2022) for any additional data or information.

**Acknowledgments:** The authors would like to thank the University of Johannesburg, Department of Chemical Sciences. To my colleagues, T.T. Letsau, B. LeGrand, N. Mabena, and C. Phela, thank you for helping me with part of my laboratory work, acquiring new synthesis methods, and presentation preparation. Appreciation to all my laboratory colleagues in Msomi's Energy Research group.

**Conflicts of Interest:** The authors declare no conflict of interest.

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
