# Peer review of "Quaternized Polyethersulfone (QPES) Membrane with Imidazole Functionalized Graphene Oxide (ImGO) for Alkaline Anion Exchange Fuel Cell Application"

_sustainability, doi:10.3390/su15032209_

Round 1

Reviewer 1 Report

The authors did very interesting research. The experiments are designed carefully and data are abundant, although the IC value is not very high. 

Author Response

The authors did very interesting research. The experiments are designed carefully, and data are abundant, although the IC value is not very high.

Thank you very much. Indeed, the IC is not very high compared to other materials found in research, but it is very comparable with other found in literature and considering the IC in this study is at room temperature. Nonetheless, we are looking at methods of increasing the IC of the same kind of membrane.

Reviewer 2 Report

A novel method was used to functionalize graphene oxide and polymer at the same time to prepare composite ion exchange membranes. The author's experimental data showed good results. But there are still some points that the authors need to pay attention to improve the quality of the paper. Please see the comments below.

Page 2 line 67, Some researchers also fabricated electrospun AEMs to enhance conductivity. Please refer "Electrospun composite proton-exchange and anion-exchange membranes for fuel cells." Energies 14.20 (2021): 6709.

Page 3 line 93, There is no reference for Yang’s work

Page 3 line 98, Is 0.5% a weight percent?

Page 6 line 225, Is the cross-sectional area from dry membrane or wet membrane?

Page 6 line 229, Why not use carbon paper and hot press MEA?

Page 10 Figure 5, I suggest authors to use SEM images with the same scale bar

Page 10 line 318, How do SEM images show that Im is uniformly dispersed in PES membrane? What are the characteristics that mark the uniform dispersion of Im? It's hard for me to understand what the author means of ‘cloudy like appearance’

Page 12 line 383, The difference between 0.5wt% and 1wt% seems too large. Authors can try 0.75wt% to check where is the limit case

Page 13 Figure 9, If the membrane can achieve 500% water absorption at 80C, then I doubt its mechanical integrity. Remember that Nafion has a water absorption of about 35%. I highly doubt that authors did not completely dry the surface when measuring water absorption

Page 13 line 395, Where are Figure 9A and 9B?

Page 13 Figure 10, The Y-axis contact Angle has no units

Page 14 line 409, Even if ImGO is poorly dispersed in the membrane, it should not affect the IEC, but the conductivity

Page 16 Figure 12, There are two 0.1% in this Figure

Page 17 Figure 13, Could authors explain why there is no polarization loss and mass transport loss in the polarization curve?

Page 17 Table 3, There are two errors in the temperature column

Author Response

Please see the attachment for corrections. 

Reviewer 3 Report

In this study, a general method is employed to improve the overall performance and properties of alkaline anion exchange membranes  by making use of polyethersulfone which is quaternized with imidazolium groups and blended with imidazolium functionalized graphene oxide inorganic filler (ImGO). The following changes and explanations should be made before it can be accepted by Sustainability.

1. The authors may clarify  the applications and novelty of this work in the  introduction of this manuscript.

2.  Are the formulas in the manuscript original or quoted? If quoted, please give the relevant references. At the same time, it is suggested to number the formulas in the manuscript.

3. The main contribution of the manuscript may be clarified in the conclusion section.

4. Some formatting errors in the manuscript need to be corrected, such as first line indentation and alignment, etc, as can be seen in sections 2.1, 2.4 and 4.4.

Author Response

In this study, a general method is employed to improve the overall performance and properties of alkaline anion exchange membranes by making use of polyethersulfone which is quaternized with imidazolium groups and blended with imidazolium functionalized graphene oxide inorganic filler (ImGO). The following changes and explanations should be made before it can be accepted by Sustainability.

  1. The authors may clarify the applications and novelty of this work in the introduction of this manuscript.

The sentence “This combination of composite from best of our knowledge it’s the first time being re-ported and evaluated in the application of methanol alkaline fuel cell”

  1. Are the formulas in the manuscript original or quoted? If quoted, please give the relevant references. At the same time, it is suggested to number the formulas in the manuscript.

The formulas are general formulas in the field, they now have been referenced and numbered. Thank you.

  1. The main contribution of the manuscript may be clarified in the conclusion section.

The conclusion has been edited and also addition the sentence “To the best of our knowledge it is the first time a membrane (PES) and GO both functionalised with 2-methyl imidazole have been reported and applied in alkaline fuel cell. The fabricated membrane adds to the body of knowledge of different composite membranes which have a potential long-term application in alkaline fuel cell” has been added on the conclusion”

  1. Some formatting errors in the manuscript need to be corrected, such as first line indentation and alignment, etc, as can be seen in sections 2.1, 2.4 and 4.4

Formatting and alignment have been revised and corrected.

Round 2

Reviewer 2 Report

Authors modified the article as suggested, and my suggestion is to accept it.